# A Structured Prediction Approach for Label Ranking

**Anna Korba, Alexandre Garcia, Florence d'Alché-Buc**
LTCI, Télécom ParisTech
Université Paris-Saclay
Paris, France
`firstname.lastname@telecom-paristech.fr`

## Abstract

We propose to solve a label ranking problem as a structured output regression task. In this view, we adopt a least square surrogate loss approach that solves a supervised learning problem in two steps: a regression step in a well-chosen feature space and a pre-image (or decoding) step. We use specific feature maps/embeddings for ranking data, which convert any ranking/permutation into a vector representation. These embeddings are all well-tailored for our approach, either by resulting in consistent estimators, or by solving trivially the pre-image problem which is often the bottleneck in structured prediction. Their extension to the case of incomplete or partial rankings is also discussed. Finally, we provide empirical results on synthetic and real-world datasets showing the relevance of our method.

## 1 Introduction

Label ranking is a prediction task which aims at mapping input instances to a (total) order over a given set of labels indexed by $\{1, \ldots, K\}$. This problem is motivated by applications where the output reflects some preferences, or order of relevance, among a set of objects. Hence there is an increasing number of practical applications of this problem in the machine learning litterature. In pattern recognition for instance (Geng and Luo, 2014), label ranking can be used to predict the different objects which are the more likely to appear in an image among a predefined set. Similarly, in sentiment analysis, (Wang et al., 2011) where the prediction of the emotions expressed in a document is cast as a label ranking problem over a set of possible affective expressions. In ad targeting, the prediction of preferences of a web user over ad categories (Djuric et al., 2014) can be also formalized as a label ranking problem, and the prediction as a ranking guarantees that each user is qualified into several categories, eliminating overexposure. Another application is metalearning, where the goal is to rank a set of algorithms according to their suitability based on the characteristics of a target dataset and learning problem (see Brazdil et al. (2003); Aiguzhinov et al. (2010)). Interestingly, the label ranking problem can also be seen as an extension of several supervised tasks, such as multiclass classification or multi-label ranking (see Dekel et al. (2004); Fürnkranz and Hüllermeier (2003)). Indeed for these tasks, a prediction can be obtained by postprocessing the output of a label ranking model in a suitable way. However, label ranking differs from other ranking problems, such as in information retrieval or recommender systems, where the goal is (generally) to predict a target variable under the form of a rating or a relevance score (Cao et al., 2007).

More formally, the goal of label ranking is to map a vector $x$ lying in some feature space $\mathcal{X}$ to a ranking $y$ lying in the space of rankings $\mathcal{Y}$. A ranking is an ordered list of items of the set $\{1, \ldots, K\}$. These relations linking the components of the $y$ objects induce a structure on the output space $\mathcal{Y}$. The label ranking task thus naturally enters the framework of structured output prediction for which an abundant litterature is available (Nowozin and Lampert, 2011). In this paper, we adopt the Surrogate Least Square Loss approach introduced in the context of output kernels (Cortes et al., 2005; Kadri et al., 2013; Brouard et al., 2016) and recently theoretically studied by Ciliberto et al.

(2016) and Osokin et al. (2017) using Calibration theory (Steinwart and Christmann, 2008). This approach divides the learning task in two steps: the first one is a vector regression step in a Hilbert space where the outputs objects are represented through an embedding, and the second one solves a pre-image problem to retrieve an output object in the $\mathcal{Y}$ space. In this framework, the algorithmic complexity of the learning and prediction tasks as well as the generalization properties of the resulting predictor crucially rely on some properties of the embedding. In this work we study and discuss some embeddings dedicated to ranking data.

Our contribution is three folds: (1) we cast the label ranking problem into the structured prediction framework and propose embeddings dedicated to ranking representation, (2) for each embedding we propose a solution to the pre-image problem and study its algorithmic complexity and (3) we provide theoretical and empirical evidence for the relevance of our method.

The paper is organized as follows. In section 2, definitions and notations of objects considered through the paper are introduced, and section 3 is devoted to the statistical setting of the learning problem. section 4 describes at length the embeddings we propose and section 5 details the theoretical and computational advantages of our approach. Finally section 6 contains empirical results on benchmark datasets.

## 2 Preliminaries

### 2.1 Mathematical background and notations

Consider a set of items indexed by $\{1, \ldots, K\}$, that we will denote $[\![K]\!]$. Rankings, i.e. ordered lists of items of $[\![K]\!]$, can be complete (i.e, involving all the items) or incomplete and for both cases, they can be without-ties (total order) or with-ties (weak order). A *full ranking* is a complete, and without-ties ranking of the items in $[\![K]\!]$. It can be seen as a permutation, i.e a bijection $\sigma : [\![K]\!] \to [\![K]\!]$, mapping each item $i$ to its rank $\sigma(i)$. The rank of item $i$ is thus $\sigma(i)$ and the item ranked at position $j$ is $\sigma^{-1}(j)$. We say that $i$ is preferred over $j$ (denoted by $i \succ j$) according to $\sigma$ if and only if $i$ is ranked lower than $j$: $\sigma(i) < \sigma(j)$. The set of all permutations over $K$ items is the symmetric group which we denote by $\mathfrak{S}_K$. A *partial ranking* is a complete ranking including ties, and is also referred as a weak order or bucket order in the litterature (see Kenkre et al. (2011)). This includes in particular the top-$k$ rankings, that is to say partial rankings dividing items in two groups, the first one being the $k \leq K$ most relevant items and the second one including all the rest. These top-k rankings are given a lot of attention because of their relevance for modern applications, especially search engines or recommendation systems (see Ailon (2010)). An *incomplete ranking* is a strict order involving only a small subset of items, and includes as a particular case pairwise comparisons, another kind of ranking which is very relevant in large-scale settings when the number of items to be ranked is very large. We now introduce the main notations used through the paper. For any function $f$, $Im(f)$ denotes the image of $f$, and $f^{-1}$ its inverse. The indicator function of any event $\mathcal{E}$ is denoted by $\mathbb{I}\{\mathcal{E}\}$. We will denote by $sign$ the function such that for any $x \in \mathbb{R}$, $sign(x) = \mathbb{I}\{x > 0\} - \mathbb{I}\{x < 0\}$. The notations $\|.\|$ and $|.|$ denote respectively the usual $l_2$ and $l_1$ norm in an Euclidean space. Finally, for any integers $a \leq b$, $[\![a, b]\!]$ denotes the set $\{a, a+1, \ldots, b\}$, and for any finite set $C$, $\#C$ denotes its cardinality.

### 2.2 Related work

An overview of label ranking algorithms can be found in Vembu and Gärtner (2010), Zhou et al. (2014)), but we recall here the main contributions. One of the first proposed approaches, called *pairwise classification* (see Fürnkranz and Hüllermeier (2003)) transforms the label ranking problem into $K(K-1)/2$ binary classification problems. For each possible pair of labels $1 \leq i < j \leq K$, the authors learn a model $m_{ij}$ that decides for any given example whether $i \succ j$ or $j \succ i$ holds. The model is trained with all examples for which either $i \succ j$ or $j \succ i$ is known (all examples for which nothing is known about this pair are ignored). At prediction time, an example is submitted to all $K(K-1)/2$ classifiers, and each prediction is interpreted as a vote for a label: if the classifier $m_{ij}$ predicts $i \succ j$, this counts as a vote for label $i$. The labels are then ranked according to the number of votes. Another approach (see Dekel et al. (2004)) consists in learning for each label a linear utility function from which the ranking is deduced. Then, a large part of the dedicated literature was devoted to adapting classical partitioning methods such as k-nearest neighbors (see Zhang and Zhou (2007), Chiang et al. (2012)) or tree-based methods, in a parametric (Cheng et al. (2010), Cheng et al.

(2009), Aledo et al. (2017)) or a non-parametric way (see Cheng and Hüllermeier (2013), Yu et al. (2010), Zhou and Qiu (2016), Clémençon et al. (2017), Sá et al. (2017)). Finally, some approaches are rule-based (see Gurrieri et al. (2012), de Sá et al. (2018)). We will compare our numerical results with the best performances attained by these methods on a set of benchmark datasets of the label ranking problem in section 6.

## 3 Structured prediction for label ranking

### 3.1 Learning problem

Our goal is to learn a function $s : \mathcal{X} \to \mathcal{Y}$ between a feature space $\mathcal{X}$ and a structured output space $\mathcal{Y}$, that we set to be $\mathfrak{S}_K$ the space of full rankings over the set of items $[\![K]\!]$. The quality of a prediction $s(x)$ is measured using a loss function $\Delta : \mathfrak{S}_K \times \mathfrak{S}_K \to \mathbb{R}$, where $\Delta(s(x), \sigma)$ is the cost suffered by predicting $s(x)$ for the true output $\sigma$. We suppose that the input/output pairs $(x, \sigma)$ come from some fixed distribution $P$ on $\mathcal{X} \times \mathfrak{S}_K$. The label ranking problem is then defined as:

$$\text{minimize}_{s:\mathcal{X} \to \mathfrak{S}_K} \mathcal{E}(s), \quad \text{with} \quad \mathcal{E}(s) = \int_{\mathcal{X} \times \mathfrak{S}_K} \Delta(s(x), \sigma) dP(x, \sigma). \tag{1}$$

In this paper, we propose to study how to solve this problem and its empirical counterpart for a family of loss functions based on some ranking embedding $\phi : \mathfrak{S}_K \to \mathcal{F}$ that maps the permutations $\sigma \in \mathfrak{S}_K$ into a Hilbert space $\mathcal{F}$:

$$\Delta(\sigma, \sigma') = \|\phi(\sigma) - \phi(\sigma')\|_{\mathcal{F}}^2. \tag{2}$$

This loss presents two main advantages: first, there exists popular losses for ranking data that can take this form within a finite dimensional Hilbert Space $\mathcal{F}$, second, this choice benefits from the theoretical results on Surrogate Least Square problems for structured prediction using Calibration Theory of Ciliberto et al. (2016) and of works of Brouard et al. (2016) on Structured Output Prediction within vector-valued Reproducing Kernel Hilbert Spaces. These works approach Structured Output Prediction along a common angle by introducing a surrogate problem involving a function $g : \mathcal{X} \to \mathcal{F}$ (with values in $\mathcal{F}$) and a surrogate loss $L(g(x), \sigma)$ to be minimized instead of Eq. 1. The surrogate loss is said to be calibrated if a minimizer for the surrogate loss is always optimal for the true loss (Calauzenes et al., 2012). In the context of true risk minimization, the surrogate problem for our case writes as:

$$\text{minimize }_{g:\mathcal{X} \to \mathcal{F}} \mathcal{R}(g), \quad \text{with} \quad \mathcal{R}(g) = \int_{\mathcal{X} \times \mathfrak{S}_K} L(g(x), \phi(\sigma)) dP(x, \sigma). \tag{3}$$

with the following surrogate loss:

$$L(g(x), \phi(\sigma)) = \|g(x) - \phi(\sigma)\|_{\mathcal{F}}^2. \tag{4}$$

Problem of Eq. (3) is in general easier to optimize since $g$ has values in $\mathcal{F}$ instead of the set of structured objects $\mathcal{Y}$, here $\mathfrak{S}_K$. The solution of (3), denoted as $g^*$, can be written for any $x \in \mathcal{X}$: $g^*(x) = \mathbb{E}[\phi(\sigma)|x]$. Eventually, a candidate $s(x)$ pre-image for $g^*(x)$ can then be obtained by solving:

$$s(x) = \underset{\sigma \in \mathfrak{S}_K}{\operatorname{argmin}} L(g^*(x), \phi(\sigma)). \tag{5}$$

In the context of Empirical Risk Minimization, a training sample $\mathcal{S} = \{(x_i, \sigma_i), i = 1, \dots, N\}$, with $N$ i.i.d. copies of the random variable $(x, \sigma)$ is available. The Surrogate Least Square approach for Label Ranking Prediction decomposes into two steps:

- Step 1: minimize a regularized empirical risk to provide an estimator of the minimizer of the regression problem in Eq. (3):

$$\text{minimize }_{g \in \mathcal{H}} \mathcal{R}_{\mathcal{S}}(g), \quad \text{with} \quad \mathcal{R}_{\mathcal{S}}(g) = \frac{1}{N} \sum_{i=1}^{N} L(g(x_i), \phi(\sigma_i)) + \Omega(g). \tag{6}$$

with an appropriate choice of hypothesis space $\mathcal{H}$ and complexity term $\Omega(g)$. We denote by $\widehat{g}$ a solution of (6).

- Step 2: solve, for any $x$ in $\mathcal{X}$, the pre-image problem that provides a prediction in the original space $\mathfrak{S}_K$:

$$\widehat{s}(x) = \operatorname*{argmin}_{\sigma \in \mathfrak{S}_K} \|\phi(\sigma) - \widehat{g}(x)\|_{\mathcal{F}}^2. \tag{7}$$

The pre-image operation can be written as $\widehat{s}(x) = d \circ \widehat{g}(x)$ with $d$ the decoding function:

$$d(h) = \operatorname*{argmin}_{\sigma \in \mathfrak{S}_K} \|\phi(\sigma) - h\|_{\mathcal{F}}^2 \text{ for all } h \in \mathcal{F}, \tag{8}$$

applied on $\widehat{g}$ for any $x \in \mathcal{X}$.

This paper studies how to leverage the choice of the embedding $\phi$ to obtain a good compromise between computational complexity and theoretical guarantees. Typically, the pre-image problem on the discrete set $\mathfrak{S}_K$ (of cardinality $K!$) can be eased for appropriate choices of $\phi$ as we show in section 4, leading to efficient solutions. In the same time, one would like to benefit from theoretical guarantees and control the excess risk of the proposed predictor $\widehat{s}$.

In the following subsection we exhibit popular losses for ranking data that we will use for the label ranking problem.

### 3.2 Losses for ranking

We now present losses $\Delta$ on $\mathfrak{S}_K$ that we will consider for the label ranking task. A natural loss for full rankings, i.e. permutations in $\mathfrak{S}_K$, is a distance between permutations. Several distances on $\mathfrak{S}_K$ are widely used in the literature (Deza and Deza, 2009), one of the most popular being the *Kendall's $\tau$ distance*, which counts the number of pairwise disagreements between two permutations $\sigma, \sigma' \in \mathfrak{S}_K$:

$$\Delta_\tau(\sigma, \sigma') = \sum_{i<j} \mathbb{I}[(\sigma(i) - \sigma(j))(\sigma'(i) - \sigma'(j)) < 0]. \tag{9}$$

The maximal Kendall's $\tau$ distance is thus $K(K-1)/2$, the total number of pairs. Another well-spread distance between permutations is the *Hamming distance*, which counts the number of entries on which two permutations $\sigma, \sigma' \in \mathfrak{S}_K$ disagree:

$$\Delta_H(\sigma, \sigma') = \sum_{i=1}^{K} \mathbb{I}[\sigma(i) \neq \sigma'(i)]. \tag{10}$$

The maximal Hamming distance is thus $K$, the number of labels or items.

The Kendall's $\tau$ distance is a natural discrepancy measure when permutations are interpreted as rankings and is thus the most widely used in the preference learning literature. In contrast, the Hamming distance is particularly used when permutations represent matching of bipartite graphs and is thus also very popular (see Fathony et al. (2018)). In the next section we show how these distances can be written as Eq. (2) for a well chosen embedding $\phi$.

## 4 Output embeddings for rankings

In what follows, we study three embeddings tailored to represent full rankings/permutations in $\mathfrak{S}_K$ and discuss their properties in terms of link with the ranking distances $\Delta_\tau$ and $\Delta_H$, and in terms of algorithmic complexity for the pre-image problem (5) induced.

### 4.1 The Kemeny embedding

Motivated by the minimization of the Kendall's $\tau$ distance $\Delta_\tau$, we study the Kemeny embedding, previously introduced for the ranking aggregation problem (see Jiao et al. (2016)):

$$\phi_\tau \colon \mathfrak{S}_K \to \mathbb{R}^{K(K-1)/2}$$
$$\sigma \mapsto (\operatorname{sign}(\sigma(j) - \sigma(i)))_{1 \le i < j \le K} .$$

which maps any permutation $\sigma \in \mathfrak{S}_K$ into $Im(\phi_\tau) \subsetneq \{-1, 1\}^{K(K-1)/2}$ (that we have embedded into the Hilbert space $(\mathbb{R}^{K(K-1)/2}, \langle ., . \rangle)$). One can show that the square of the euclidean distance

between the mappings of two permutations $\sigma, \sigma' \in \mathfrak{S}_K$ recovers their Kendall's $\tau$ distance (proving at the same time that $\phi_\tau$ is injective) up to a constant: $\|\phi_\tau(\sigma) - \phi_\tau(\sigma')\|^2 = 4\Delta_\tau(\sigma, \sigma')$. The Kemeny embedding then naturally appears to be a good candidate to build a surrogate loss related to $\Delta_\tau$. By noticing that $\phi_\tau$ has a constant norm ($\forall \sigma \in \mathfrak{S}_K, \|\phi_\tau(\sigma)\| = \sqrt{K(K-1)/2}$), we can rewrite the pre-image problem (7) under the form:

$$\widehat{s}(x) = \underset{\sigma \in \mathfrak{S}_K}{\operatorname{argmin}} -\langle \phi_\tau(\sigma), \widehat{g}(x) \rangle. \tag{11}$$

To compute (11), one can first solve an Integer Linear Program (ILP) to find $\widehat{\phi_\sigma} = \operatorname{argmin}_{\phi_\sigma \in Im(\phi_\tau)} -\langle \phi_\sigma, \widehat{g}(x) \rangle$, and then find the output object $\sigma = \phi_\tau^{-1}(\widehat{\phi_\sigma})$. The latter step, i.e. inverting $\phi_\tau$, can be performed in $\mathcal{O}(K^2)$ by means of the Copeland method (see Merlin and Saari (1997)), which ranks the items by their number of pairwise victories[1]. In contrast, the ILP problem is harder to solve since it involves a minimization over $Im(\phi_\tau)$, a set of structured vectors since their coordinates are strongly correlated by the *transitivity* property of rankings. Indeed, consider a vector $v \in Im(\phi_\tau)$, so $\exists \sigma \in \mathfrak{S}_K$ such that $v = \phi_\tau(\sigma)$. Then, for any $1 \le i < j < k \le K$, if its coordinates corresponding to the pairs $(i, j)$ and $(j, k)$ are equal to one (meaning that $\sigma(i) < \sigma(j)$ and $\sigma(j) < \sigma(k)$), then the coordinate corresponding to the pair $(i, k)$ cannot contradict the others and must be set to one as well. Since $\phi_\sigma = (\phi_\sigma)_{i,j} \in Im(\phi_\tau)$ is only defined for $1 \le i < j \le K$, one cannot directly encode the transitivity constraints that take into account the components $(\phi_\sigma)_{i,j}$ with $j > i$. Thus to encode the transitivity constraint we introduce $\phi'_\sigma = (\phi'_\sigma)_{i,j} \in \mathbb{R}^{K(K-1)}$ defined by $(\phi'_\sigma)_{i,j} = (\phi_\sigma)_{i,j}$ if $1 \le i < j \le K$ and $(\phi'_\sigma)_{i,j} = -(\phi_\sigma)_{i,j}$ else, and write the ILP problem as follows:

$$\widehat{\phi_\sigma} = \underset{\phi'_\sigma}{\operatorname{argmin}} \sum_{1 \le i,j \le K} \widehat{g}(x)_{i,j}(\phi'_\sigma)_{i,j},$$
$$s.c. \begin{cases} (\phi'_\sigma)_{i,j} \in \{-1, 1\} & \forall i, j \\ (\phi'_\sigma)_{i,j} + (\phi'_\sigma)_{j,i} = 0 & \forall i, j \\ -1 \le (\phi'_\sigma)_{i,j} + (\phi'_\sigma)_{j,k} + (\phi'_\sigma)_{k,i} \le 1 & \forall i, j, k \text{ s.t. } i \neq j \neq k. \end{cases} \tag{12}$$

Such a problem is NP-Hard. In previous works (see Calauzenes et al. (2012); Ramaswamy et al. (2013)), the complexity of designing calibrated surrogate losses for the Kendall's $\tau$ distance had already been investigated. In particular, Calauzenes et al. (2012) proved that there exists no convex $K$-dimensional calibrated surrogate loss for Kendall's $\tau$ distance. As a consequence, optimizing this type of loss has an inherent computational cost. However, in practice, branch and bound based ILP solvers find the solution of (12) in a reasonable time for a reduced number of labels $K$. We discuss the computational implications of choosing the Kemeny embedding section 5.2. We now turn to the study of an embedding devoted to build a surrogate loss for the Hamming distance.

## 4.2 The Hamming embedding

Another well-spread embedding for permutations, that we will call the Hamming embedding, consists in mapping $\sigma$ to its permutation matrix $\phi_H(\sigma)$:

$$\phi_H : \mathfrak{S}_K \to \mathbb{R}^{K \times K}$$
$$\sigma \mapsto (\mathbb{I}\{\sigma(i) = j\})_{1 \le i,j \le K},$$

where we have embedded the set of permutation matrices $Im(\phi_H) \subsetneq \{0, 1\}^{K \times K}$ into the Hilbert space $(\mathbb{R}^{K \times K}, \langle ., . \rangle)$ with $\langle ., . \rangle$ the Froebenius inner product. This embedding shares similar properties with the Kemeny embedding: first, it is also of constant (Froebenius) norm, since $\forall \sigma \in \mathfrak{S}_K, \|\phi_H(\sigma)\| = \sqrt{K}$. Then, the squared euclidean distance between the mappings of two permutations $\sigma, \sigma' \in \mathfrak{S}_K$ recovers their Hamming distance (proving that $\phi_H$ is also injective): $\|\phi_H(\sigma) - \phi_H(\sigma')\|^2 = \Delta_H(\sigma, \sigma')$. Once again, the pre-image problem consists in solving the linear program:

$$\widehat{s}(x) = \underset{\sigma \in \mathfrak{S}_K}{\operatorname{argmin}} -\langle \phi_H(\sigma), \widehat{g}(x) \rangle, \tag{13}$$

which is, as for the Kemeny embedding previously, divided in a minimization step, i.e. find $\widehat{\phi_\sigma} = \text{argmin}_{\phi_\sigma \in Im(\phi_H)} -\langle \phi_\sigma, g(x) \rangle$, and an inversion step, i.e. compute $\sigma = \phi_H^{-1}(\widehat{\phi_\sigma})$. The inversion step is of complexity $\mathcal{O}(K^2)$ since it involves scrolling through all the rows (items $i$) of the matrix $\widehat{\phi_\sigma}$ and all the columns (to find their positions $\sigma(i)$). The minimization step itself writes as the following problem:

$$\widehat{\phi_\sigma} = \underset{\phi_\sigma}{\text{argmax}} \sum_{1 \leq i,j \leq K} \widehat{g}(x)_{i,j}(\phi_\sigma)_{i,j},$$

$$s.c \begin{cases} (\phi_\sigma)_{i,j} \in \{0,1\} & \forall\, i,j \\ \sum_i (\phi_\sigma)_{i,j} = \sum_j (\phi_\sigma)_{i,j} = 1 & \forall\, i,j\,, \end{cases} \tag{14}$$

which can be solved with the Hungarian algorithm (see Kuhn (1955)) in $\mathcal{O}(K^3)$ time. Now we turn to the study of an embedding which presents efficient algorithmic properties.

### 4.3 Lehmer code

A permutation $\sigma = (\sigma(1), \ldots, \sigma(K)) \in \mathfrak{S}_K$ may be uniquely represented via its Lehmer code (also called the inversion vector), i.e. a word of the form $c_\sigma \in \mathcal{C}_K \triangleq \{0\} \times [\![0,1]\!] \times [\![0,2]\!] \times \cdots \times [\![0, K-1]\!]$, where for $j = 1, \ldots, K$:

$$c_\sigma(j) = \#\{i \in [\![K]\!] : i < j, \sigma(i) > \sigma(j)\}. \tag{15}$$

The coordinate $c_\sigma(j)$ is thus the number of elements $i$ with index smaller than $j$ that are ranked higher than $j$ in the permutation $\sigma$. By default, $c_\sigma(1) = 0$ and is typically omitted. For instance, we have:

| e | 1 | 2 | 3 | 4 | 5 | 6 | 7 | 8 | 9 |
|---|---|---|---|---|---|---|---|---|---|
| $\sigma$ | 2 | 1 | 4 | 5 | 7 | 3 | 6 | 9 | 8 |
| $c_\sigma$ | 0 | 1 | 0 | 0 | 0 | 3 | 1 | 0 | 1 |

It is well known that the Lehmer code is bijective, and that the encoding and decoding algorithms have linear complexity $\mathcal{O}(K)$ (see Mareš and Straka (2007), Myrvold and Ruskey (2001)). This embedding has been recently used for ranking aggregation of full or partial rankings (see Li et al. (2017)). Our idea is thus to consider the following Lehmer mapping for label ranking;

$$\phi_L : \mathfrak{S}_K \to \mathbb{R}^K$$
$$\sigma \mapsto (c_\sigma(i))_{i=1,\ldots,K}\,,$$

which maps any permutation $\sigma \in \mathfrak{S}_K$ into the space $\mathcal{C}_K$ (that we have embedded into the Hilbert space $(\mathbb{R}^K, \langle ., . \rangle)$). The loss function in the case of the Lehmer embedding is thus the following:

$$\Delta_L(\sigma, \sigma') = \|\phi_L(\sigma) - \phi_L(\sigma')\|^2, \tag{16}$$

which does not correspond to a known distance over permutations (Deza and Deza, 2009). Notice that $|\phi_L(\sigma)| = d_\tau(\sigma, e)$ where $e$ is the identity permutation, a quantity which is also called the number of inversions of $\sigma$. Therefore, in contrast to the previous mappings, the norm $\|\phi_L(\sigma)\|$ is not constant for any $\sigma \in \mathfrak{S}_K$. Hence it is not possible to write the loss $\Delta_L(\sigma, \sigma')$ as $-\langle \phi_L(\sigma), \phi_L(\sigma') \rangle$[2]. Moreover, this mapping is not distance preserving and it can be proven that $\frac{1}{K-1}\Delta_\tau(\sigma, \sigma') \leq |\phi_L(\sigma) - \phi_L(\sigma')| \leq \Delta_\tau(\sigma, \sigma')$ (see Wang et al. (2015)). However, the Lehmer embedding still enjoys great advantages. Firstly, its coordinates are decoupled, which will enable a trivial solving of the inverse image step (7). Indeed we can write explicitly its solution as:

$$\widehat{s}(x) = \underbrace{\phi_L^{-1} \circ d_L}_{d} \circ \widehat{g}(x) \quad \text{with} \quad \begin{aligned} d_L : &\mathbb{R}^K \to \mathcal{C}_K \\ &(h_i)_{i=1,\ldots,K} \mapsto (\underset{j \in [\![0, i-1]\!]}{\text{argmin}} (h_i - j))_{i=1,\ldots,K}, \end{aligned} \tag{17}$$

where $d$ is the decoding function defined in (8). Then, there may be repetitions in the coordinates of the Lehmer embedding, allowing for a compact representation of the vectors.

## 4.4 Extension to partial and incomplete rankings

In many real-world applications, one does not observe full rankings but only partial or incomplete rankings (see the definitions section 2.1). We now discuss to what extent the embeddings we propose for permutations can be adapted to this kind of rankings *as input data*. Firstly, the Kemeny embedding can be naturally extended to partial and incomplete rankings since it encodes *relative* information about the positions of the items. Indeed, we propose to map any partial ranking $\widetilde{\sigma}$ to the vector:

$$\phi(\widetilde{\sigma}) = (sign(\widetilde{\sigma}(i) - \widetilde{\sigma}(j))_{1 \leq i < j \leq K}, \tag{18}$$

where each coordinate can now take its value in $\{-1, 0, 1\}$ (instead of $\{-1, 1\}$ for full rankings). For any incomplete ranking $\bar{\sigma}$, we also propose to fill the missing entries (missing comparisons) in the embedding with zeros. This can be interpreted as setting the probability that $i \succ j$ to 1/2 for a missing comparison between $(i, j)$. In contrast, the Hamming embedding, since it encodes the absolute positions of the items, is tricky to extend to map partial or incomplete rankings where this information is missing. Finally, the Lehmer embedding falls between the two latter embeddings. It also relies on an encoding of relative rankings and thus may be adapted to take into account the partial ranking information. Indeed, in Li et al. (2017), the authors propose a generalization of the Lehmer code for partial rankings. We recall that a tie in a ranking happens when $\#\{i \neq j, \sigma(i) = \sigma(j)\} > 0$. The generalized representation $c'$ takes into account ties, so that for any partial ranking $\widetilde{\sigma}$:

$$c'_{\widetilde{\sigma}}(j) = \#\{i \in [\![K]\!] : i < j, \widetilde{\sigma}(i) \geq \widetilde{\sigma}(j)\}. \tag{19}$$

Clearly, $c'_{\widetilde{\sigma}}(j) \geq c_{\widetilde{\sigma}}(j)$ for all $j \in [\![K]\!]$. Given a partial ranking $\widetilde{\sigma}$, it is possible to break its ties to convert it in a permutation $\sigma$ as follows: for $i, j \in [\![K]\!]^2$, if $\widetilde{\sigma}(i) = \widetilde{\sigma}(j)$ then $\sigma(i) = \sigma(j)$ iff $i < j$. The entries $j = 1, \ldots, K$ of the Lehmer codes of $\widetilde{\sigma}$ (see (20)) and $\sigma$ (see (15)) then verify:

$$c'_{\widetilde{\sigma}}(j) = c_\sigma(j) + IN_j - 1 \quad , \quad c_{\widetilde{\sigma}}(j) = c_\sigma(j), \tag{20}$$

where $IN_j = \#\{i \leq j, \widetilde{\sigma}(i) = \widetilde{\sigma}(j)\}$. An example illustrating the extension of the Lehmer code to partial rankings is given in the Supplementary. However, computing each coordinate of the Lehmer code $c_\sigma(j)$ for any $j \in [\![K]\!]$ requires to sum over the $[\![K]\!]$ items. As an incomplete ranking do not involve the whole set of items, it is also tricky to extend the Lehmer code to map incomplete rankings.

Taking as input partial or incomplete rankings only modifies Step 1 of our method since it corresponds to the mapping step of the training data, and in Step 2 we still predict a full ranking. Extending our method to the task of predicting as output a partial or incomplete ranking raises several mathematical questions that we did not develop at length here because of space limitations. For instance, to predict partial rankings, a naive approach would consist in predicting a full ranking and then converting it to a partial ranking according to some threshold (i.e, keep the top-k items of the full ranking). A more formal extension of our method to make it able to predict directly partial rankings as outputs would require to optimize a metric tailored for this data and which could be written as in Eq. (2). A possibility for future work could be to consider the extension of the Kendall's $\tau$ distance with penalty parameter $p$ for partial rankings proposed in Fagin et al. (2004).

## 5 Computational and theoretical analysis

### 5.1 Theoretical guarantees

In this section, we give some statistical guarantees for the estimators obtained by following the steps described in section 3. To this end, we build upon recent results in the framework of Surrogate Least Square by Ciliberto et al. (2016). Consider one of the embeddings $\phi$ on permutations presented in the previous section, which defines a loss $\Delta$ as in Eq. (2). Let $c_\phi = \max_{\sigma \in \mathfrak{S}_K} \|\phi(\sigma)\|$. We will denote by $s^*$ a minimizer of the true risk (1), $g^*$ a minimizer of the surrogate risk (3), and $d$ a decoding function as (8)[3]. Given an estimator $\widehat{g}$ of $g^*$ from Step 1, i.e. a minimizer of the empirical surrogate risk (6) we can then consider in Step 2 an estimator $\widehat{s} = d \circ \widehat{g}$. The following theorem reveals how the performance of the estimator $\widehat{s}$ we propose can be related to a solution $s^*$ of (1) for the considered embeddings.

| Embedding | Step 1 (a) | Step 2 (b) | | Regressor | Step 1 (b) | Step 2 (a) |
|---|---|---|---|---|---|---|
| $\phi_\tau$ | $\mathcal{O}(K^2N)$ | NP-hard | | kNN | $\mathcal{O}(1)$ | $\mathcal{O}(Nm)$ |
| $\phi_H$ | $\mathcal{O}(KN)$ | $\mathcal{O}(K^3N)$ | | Ridge | $\mathcal{O}(N^3)$ | $\mathcal{O}(Nm)$ |
| $\phi_L$ | $\mathcal{O}(KN)$ | $\mathcal{O}(KN)$ | | | | |

Table 1: Embeddings and regressors complexities.

**Theorem 1** *The excess risks of the proposed predictors are linked to the excess surrogate risks as:*

*(i) For the loss (2) defined by the Kemeny and Hamming embedding $\phi_\tau$ and $\phi_H$ respectively:*

$$\mathcal{E}(d \circ \widehat{g}) - \mathcal{E}(s^*) \leq c_\phi \sqrt{\mathcal{R}(\widehat{g}) - \mathcal{R}(g^*)}$$

*with $c_{\phi_\tau} = \sqrt{\frac{K(K-1)}{2}}$ and $c_{\phi_H} = \sqrt{K}$.*

*(ii) For the loss (2) defined by the Lehmer embedding $\phi_L$:*

$$\mathcal{E}(d \circ \widehat{g}) - \mathcal{E}(s^*) \leq \sqrt{\frac{K(K-1)}{2}} \sqrt{\mathcal{R}(\widehat{g}) - \mathcal{R}(g^*)} + \mathcal{E}(d \circ g^*) - \mathcal{E}(s^*) + \mathcal{O}(K\sqrt{K})$$

The full proof is given in the Supplementary. Assertion (i) is a direct application of Theorem 2 in Ciliberto et al. (2016). In particular, it comes from a preliminary consistency result which shows that $\mathcal{E}(d \circ g^*) = \mathcal{E}(s^*)$ for both embeddings. Concerning the Lehmer embedding, it is not possible to apply their consistency results immediately; however a large part of the arguments of their proof is used to bound the estimation error for the surrogate risk, and we remain with an approximation error $\mathcal{E}(d \circ g^*) - \mathcal{E}(s^*) + \mathcal{O}(K\sqrt{K})$ resulting in Assertion (ii). In Remark 2 in the Supplementary, we give several insights about this approximation error. Firstly we show that it can be upper bounded by $2\sqrt{2}\sqrt{K(K-1)}\mathcal{E}(s^*) + \mathcal{O}(K\sqrt{K})$. Then, we explain how this term results from using $\phi_L$ in the learning procedure. The Lehmer embedding thus have weaker statistical guarantees, but has the advantage of being more computationnally efficient, as we explain in the next subsection.

Notice that for Step 1, one can choose a consistent regressor with vector values $\widehat{g}$, i.e such that $\mathcal{R}(\widehat{g}) \to \mathcal{R}(g^*)$ when the number of training points tends to infinity. Examples of such methods that we use in our experiments to learn $\widehat{g}$, are the k-nearest neighbors (kNN) or kernel ridge regression (Micchelli and Pontil, 2005) methods whose consistency have been proved (see Chapter 5 in Devroye et al. (2013) and Caponnetto and De Vito (2007)). In this case the control of the excess of the surrogate risk $\mathcal{R}(\widehat{g}) - \mathcal{R}(g^*)$ implies the control of $\mathcal{E}(\widehat{s}) - \mathcal{E}(s^*)$ where $\widehat{s} = d \circ \widehat{g}$ by Theorem 1.

**Remark 1** *We clarify that the consistency results of Theorem 1 are established for the task of predicting full rankings which is adressed in this paper. In the case of predicting partial or incomplete rankings, these results are not guaranteed to hold. Providing theoretical guarantees for this task is left for future work.*

## 5.2 Algorithmic complexity

We now discuss the algorithmic complexity of our approach. We recall that $K$ is the number of items/labels whereas $N$ is the number of samples in the dataset. For a given embedding $\phi$, the total complexity of our approach for learning decomposes as follows. Step 1 in Section 3 can be decomposed in two steps: a preprocessing step (Step 1 (a)) consisting in mapping the training sample $\{(x_i, \sigma_i), i = 1, \ldots, N\}$ to $\{(x_i, \phi(\sigma_i)), i = 1, \ldots, N\}$, and a second step (Step 1 (b)) that consists in computing the estimator $\widehat{g}$ of the Least squares surrogate empirical minimization (6). Then, at prediction time, Step 2 Section 3 can also be decomposed in two steps: a first one consisting in mapping new inputs to a Hilbert space using $\widehat{g}$ (Step 2 (a)), and then solving the preimage problem (7) (Step 2 (b)). The complexity of a predictor corresponds to the worst complexity across all steps. The complexities resulting from the choice of an embedding and a regressor are summarized Table 1, where we denoted by $m$ the dimension of the ranking embedded representations. The Lehmer embedding with kNN regressor thus provides the fastest theoretical complexity of $\mathcal{O}(KN)$ at the cost of weaker theoretical guarantees. The fastest methods previously proposed in the litterature typically involved a sorting procedure at prediction Cheng et al. (2010) leading to a $\mathcal{O}(NKlog(K))$ complexity. In the experimental section we compare our approach with the former (denoted as Cheng

PL), but also with the label wise decomposition approach in Cheng and Hüllermeier (2013) (Cheng LWD) involving a kNN regression followed by a projection on $\mathfrak{S}_K$ computed in $\mathcal{O}(K^3 N)$, and the more recent Random Forest Label Ranking (Zhou RF) Zhou and Qiu (2016). In their analysis, if $d_{\mathcal{X}}$ is the size of input features and $D_{\max}$ the maximum depth of a tree, then RF have a complexity in $\mathcal{O}(D_{\max} d_{\mathcal{X}} K^2 N^2)$.

## 6 Numerical Experiments

Finally we evaluate the performance of our approach on standard benchmarks. We present the results obtained with two regressors : Kernel Ridge regression (Ridge) and k-Nearest Neighbors (kNN). Both regressors were trained with the three embeddings presented in Section 4. We adopt the same setting as Cheng et al. (2010) and report the results of our predictors in terms of mean Kendall's $\tau$:

$$k_\tau = \frac{C - D}{K(K - 1)/2} \quad \begin{cases} C : \text{number of concordant pairs between 2 rankings} \\ D : \text{number of discordant pairs between 2 rankings} \end{cases}, \quad (21)$$

from five repetitions of a ten-fold cross-validation (c.v.). Note that $k_\tau$ is an affine transformation of the Kendall's tau distance $\Delta_\tau$ mapping on the $[-1, 1]$ interval. We also report the standard deviation of the resulting scores as in Cheng and Hüllermeier (2013). The parameters of our regressors were tuned in a five folds inner c.v. for each training set. We report our parameter grids in the supplementary materials.

Table 2: Mean Kendall's $\tau$ coefficient on benchmark datasets

|  | authorship | glass | iris | vehicle | vowel | wine |
|---|---|---|---|---|---|---|
| kNN Hamming | 0.01±0.02 | 0.08±0.04 | -0.15±0.13 | -0.21±0.04 | 0.24±0.04 | -0.36±0.04 |
| kNN Kemeny | **0.94**±0.02 | 0.85±0.06 | 0.95±0.05 | 0.85±0.03 | 0.85±0.02 | 0.94±0.05 |
| kNN Lehmer | 0.93±0.02 | 0.85±0.05 | 0.95±0.04 | 0.84±0.03 | 0.78±0.03 | 0.94±0.06 |
| ridge Hamming | -0.00±0.02 | 0.08±0.05 | -0.10±0.13 | -0.21±0.03 | 0.26±0.04 | -0.36±0.03 |
| ridge Lehmer | 0.92±0.02 | 0.83±0.05 | **0.97**±0.03 | 0.85±0.02 | 0.86±0.01 | 0.84±0.08 |
| ridge Kemeny | **0.94**±0.02 | 0.86±0.06 | **0.97**±0.05 | **0.89**±0.03 | **0.92**±0.01 | 0.94±0.05 |
| Cheng PL | **0.94**±0.02 | 0.84±0.07 | 0.96±0.04 | 0.86±0.03 | 0.85±0.02 | **0.95**±0.05 |
| Cheng LWD | 0.93±0.02 | 0.84±0.08 | 0.96±0.04 | 0.85±0.03 | 0.88±0.02 | 0.94±0.05 |
| Zhou RF | 0.91 | **0.89** | **0.97** | 0.86 | 0.87 | **0.95** |

The Kemeny and Lehmer embedding based approaches are competitive with the state of the art methods on these benchmarks datasets. The Hamming based methods give poor results in terms of $k_\tau$ but become the best choice when measuring the mean Hamming distance between predictions and ground truth (see Table 3 in the Supplementary). In contrast, the fact that the Lehmer embedding performs well for the optimization of the Kendall's $\tau$ distance highlights its practical relevance for label ranking. The Supplementary presents additional results (on additional datasets and results in terms of Hamming distance) which show that our method remains competitive with the state of the art. The code to reproduce our results is available: `https://github.com/akorba/Structured_Approach_Label_Ranking/`

## 7 Conclusion

This paper introduces a novel framework for label ranking, which is based on the theory of Surrogate Least Square problem for structured prediction. The structured prediction approach we propose comes along with theoretical guarantees and efficient algorithms, and its performance has been shown on real-world datasets. To go forward, extensions of our methodology to predict partial and incomplete rankings are to be investigated. In particular, the framework of prediction with abstention should be of interest.

## Footnotes

[1]Copeland method firstly affects a score $s_i$ for item $i$ as: $s_i = \sum_{j \neq i} \mathbb{I}\{\sigma(i) < \sigma(j)\}$ and then ranks the items by decreasing score.

[2]The scalar product of two embeddings of two permutations $\phi_L(\sigma), \phi_L(\sigma')$ is not maximized for $\sigma = \sigma'$.

[3]Note that $d = \phi_L^{-1} \circ d_L$ for $\phi_L$ and is obtained as the composition of two steps for $\phi_\tau$ and $\phi_H$: solving an optimization problem and compute the inverse of the embedding.

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
