[Supplementary Material]

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

# 8 Supplementary material

## 8.1 Proof of Theorem 1

We borrow the notations of Ciliberto et al. (2016) and recall their main result Theorem 2. They firstly exhibit the following assumption for a given loss $\Delta$, see Assumption 1 therein:

**Assumption 1.** There exists a separable Hilbert space $\mathcal{F}$ with inner product $\langle ., . \rangle_{\mathcal{F}}$, a continuous embedding $\psi : \mathcal{Y} \to \mathcal{F}$ and a bounded linear operator $V : \mathcal{F} \to \mathcal{F}$, such that:

$$\Delta(y, y') = \langle \psi(y), V\psi(y') \rangle_{\mathcal{F}} \quad \forall y, y' \in \mathcal{Y} \tag{22}$$

**Theorem 2** *Let $\Delta : \mathcal{Y} \to \mathcal{Y}$ satisfying Assumption 1 with $\mathcal{Y}$ a compact set. Then, for every measurable $g : \mathcal{X} \to \mathcal{F}$ and $d : \mathcal{F} \to \mathcal{Y}$ such that $\forall h \in \mathcal{F}$, $d(h) = \mathrm{argmin}_{y \in \mathcal{Y}} \langle \phi(y), h \rangle_{\mathcal{F}}$, the following holds:*

    *(i) Fisher Consistency: $\mathcal{E}(d \circ g^*) = \mathcal{E}(s^*)$*

    *(ii) Comparison Inequality: $\mathcal{E}(d \circ g) - \mathcal{E}(s^*) \leq 2c_\Delta \sqrt{\mathcal{R}(g) - \mathcal{R}(g^*)}$*

*with $c_\Delta = \|V\| \max_{y \in \mathcal{Y}} \|\phi(y)\|$.*

Notice that any discrete set $\mathcal{Y}$ is compact and $\phi : \mathcal{Y} \to \mathcal{F}$ is continuous. We now prove the two assertions of Theorem 1.

*Proof of Assertion(i) in Theorem 1.* Firstly, $\mathcal{Y} = \mathfrak{S}_K$ is finite. Then, for the Kemeny and Hamming embeddings, $\Delta$ satisfies Assumption 1 with $V = -id$ (where $id$ denotes the identity operator) , and $\psi = \phi_K$ and $\psi = \phi_H$ respectively. Theorem 2 thus applies directly.

*Proof of Assertion(ii) in Theorem 1.* In the following proof, $\mathcal{Y}$ denotes $\mathfrak{S}_K$, $\phi$ denotes $\phi_L$ and $d = \phi_L^{-1} \circ d_L$ with $d_L$ as defined in (17). Our goal is to control the excess risk $\mathcal{E}(s) - \mathcal{E}(s^*)$.

$$\mathcal{E}(s) - \mathcal{E}(s^*) = \mathcal{E}(d \circ \widehat{g}) - \mathcal{E}(s^*)$$
$$= \underbrace{\mathcal{E}(d \circ \widehat{g}) - \mathcal{E}(d \circ g^*)}_{(A)} + \underbrace{\mathcal{E}(d \circ g^*) - \mathcal{E}(s^*)}_{(B)}$$

Consider the first term (A).

$$\mathcal{E}(d \circ \widehat{g}) - \mathcal{E}(d \circ g^*) = \int_{\mathcal{X} \times \mathcal{Y}} \Delta(d \circ \widehat{g}(x), \sigma) - \Delta(d \circ g^*(x), \sigma) dP(x, \sigma)$$

$$= \int_{\mathcal{X} \times \mathcal{Y}} \|\phi(d \circ \widehat{g}(x)) - \phi(\sigma)\|_{\mathcal{F}}^2 - \|\phi(d \circ g^*(x)) - \phi(\sigma)\|_{\mathcal{F}}^2 dP(x, \sigma)$$

$$= \underbrace{\int_{\mathcal{X}} \|\phi(d \circ \widehat{g}(x))\|_{\mathcal{F}}^2 - \|\phi(d \circ g^*(x))\|_{\mathcal{F}}^2 dP(x) +}_{(A1)}$$

$$\underbrace{2 \int_{\mathcal{X}} \langle \phi(d \circ g^*(x)) - \phi(d \circ \widehat{g}(x)), \int_{\mathcal{Y}} \phi(\sigma) dP(\sigma, x) \rangle dP(x)}_{(A2)}$$

The first term (A1) can be upper bounded as follows:

$$\int_{\mathcal{X}} \|\phi(d \circ \widehat{g}(x))\|_{\mathcal{F}}^2 - \|\phi(d \circ g^*(x))\|_{\mathcal{F}}^2 dP(x) \leq \int_{\mathcal{X}} \langle \phi(d \circ \widehat{g}(x)) - \phi(d \circ g^*(x)), \phi(d \circ \widehat{g}(x)) + \phi(d \circ g^*(x)) \rangle_{\mathcal{F}} dP(x)$$

$$\leq 2c_\Delta \int_{\mathcal{X}} \|\phi(d \circ \widehat{g}(x)) - \phi(d \circ g^*(x))\|_{\mathcal{F}} dP(x)$$

$$\leq 2c_\Delta \sqrt{\int_{\mathcal{X}} \|d_L(\widehat{g}(x)) - d_L(g^*(x))\|_{\mathcal{F}}^2 dP(x)}$$

$$\leq 2c_\Delta \sqrt{\int_{\mathcal{X}} \|g^*(x) - \widehat{g}(x)\|_{\mathcal{F}}^2 dP(x)} + \mathcal{O}(K\sqrt{K})$$

with $c_\Delta = \max_{\sigma \in \mathcal{Y}} \|\phi(\sigma)\|_{\mathcal{F}} = \sqrt{\frac{(K-1)(K-2)}{2}}$ and since $\|d_L(u) - d_L(v)\| \leq \|u - v\| + \sqrt{K}$. Since $\int_{\mathcal{X}} \|g^*(x) - \widehat{g}(x)\|_{\mathcal{F}}^2 dP(x) = \mathcal{R}(\widehat{g}) - \mathcal{R}(g^*)$ (see Ciliberto et al. (2016)) we get the first term of Assertion (i).

For the second term (A2), we can actually follow the proof of Theorem 12 in Ciliberto et al. (2016) and we get:

$$\int_{\mathcal{X}} \langle \phi(d \circ g^*(x)) - \phi(d \circ \widehat{g}(x)), \int_{\mathcal{Y}} \phi(\sigma) dP(\sigma, x) \rangle dP(x) \leq 2c_\Delta \sqrt{\mathcal{R}(\widehat{g}) - \mathcal{R}(g^*)}$$

Consider the second term (2). By Lemma 8 in (Ciliberto et al., 2016), we have that:

$$g^*(x) = \int_{\mathcal{Y}} \phi(\sigma) dP(\sigma|x) \tag{23}$$

and then:

$$\begin{aligned}
\mathcal{E}(d \circ g^*) - \mathcal{E}(s^*) &= \int_{\mathcal{X} \times \mathcal{Y}} \|\phi(d \circ g^*(x)) - \phi(\sigma)\|_{\mathcal{F}}^2 - \|\phi(s^*(x)) - \phi(\sigma)\|_{\mathcal{F}}^2 dP(x, \sigma) \\
&\leq \int_{\mathcal{X} \times \mathcal{Y}} \langle \phi(d \circ \widehat{g}(x)) - \phi(s^*(x)), \phi(d \circ \widehat{g}(x)) + \phi(s^*(x)) - 2\phi(\sigma) \rangle_{\mathcal{F}} dP(x, \sigma) \\
&\leq 4c_\Delta \int_{\mathcal{X}} \|\phi(d \circ g^*(x)) - \phi(s^*(x))\|_{\mathcal{F}} dP(x) \\
&\leq 4c_\Delta \int_{\mathcal{X}} \|d_L \circ g^*(x)) - d_L \circ \phi(s^*(x))\|_{\mathcal{F}} dP(x) \\
&\leq 4c_\Delta \int_{\mathcal{X}} \|g^*(x)) - \phi(s^*(x))\|_{\mathcal{F}} dP(x) + \mathcal{O}(K\sqrt{K})
\end{aligned}$$

where we used that $\phi(s^*(x)) \in \mathcal{C}_K$ so $d_L \circ \phi(s^*(x)) = \phi(s^*(x))$. Then we can plug (23) in the right term:

$$\begin{aligned}
\mathcal{E}(d \circ g^*) - \mathcal{E}(s^*) &\leq 4c_\Delta \int_{\mathcal{X}} \| \int_{\mathcal{Y}} \phi(\sigma) dP(\sigma|x) - \phi(s^*(x))\|_{\mathcal{F}} dP(x) + \mathcal{O}(K\sqrt{K}) \\
&\leq 4c_\Delta \int_{\mathcal{X} \times \mathcal{Y}} \|\phi(\sigma) - \phi(s^*(x))\|_{\mathcal{F}} dP(x) + \mathcal{O}(K\sqrt{K}) \\
&\leq 4c_\Delta \mathcal{E}(s^*) + \mathcal{O}(K\sqrt{K})
\end{aligned}$$

**Remark 2** *As proved in Theorem 19 in (Ciliberto et al., 2016), since the space of rankings $\mathcal{Y}$ is finite, $\Delta_L$ necessarily satisfies Assumption 1 with some continuous embedding $\psi$. If the approach we developped was relying on this $\psi$, we would have consistency for the minimizer $g^*$ of the Lehmer loss (16). However, the choice of $\phi_L$ is relevant because it yields a pre-image problem with low computational complexity.*

## 8.2 Lehmer embedding for partial rankings

An example, borrowed from (Li et al., 2017) illustrating the extension of the Lehmer code for partial rankings is the following:

| $e$ | 1 | 2 | 3 | 4 | 5 | 6 | 7 | 8 | 9 |
|---|---|---|---|---|---|---|---|---|---|
| $\widetilde{\sigma}$ | 1 | 1 | 2 | 2 | 3 | 1 | 2 | 3 | 3 |
| $\sigma$ | 1 | 2 | 4 | 5 | 7 | 3 | 6 | 8 | 9 |
| $c_\sigma$ | 0 | 0 | 0 | 0 | 0 | 3 | 1 | 0 | 0 |
| $IN$ | 1 | 2 | 1 | 2 | 1 | 3 | 3 | 2 | 3 |
| $c_{\widetilde{\sigma}}$ | 0 | 0 | 0 | 0 | 0 | 3 | 1 | 0 | 0 |
| $c'_{\widetilde{\sigma}}$ | 0 | 1 | 0 | 1 | 0 | 5 | 3 | 1 | 2 |

where each row represents a step to encode a partial ranking.

## 8.3 Additional experimental results

**Details concerning the parameter grids.** We first recall our notations for vector valued kernel ridge regression. Let $\mathcal{H}_K$ be a vector-valued Reproducing Kernel Hilbert Space associated to an operator-valued kernel $K : \mathcal{X} \times \mathcal{X} \to \mathcal{L}(\mathbb{R}^n)$. Solve:

$$\min_{g \in \mathcal{H}_K} \sum_{k=1}^{N} \|g(x_k) - \phi(\sigma_k)\|^2 + \lambda \|h\|_{\mathcal{H}_K}^2 \tag{24}$$

The solution of this problem is unique and admits an expansion: $\widehat{g}(.) = \sum_{i=1}^{N} K(x_i, .)c_i$ (see Micchelli and Pontil (2005)). Moreover, it has the following closed-form solution:

$$\widehat{g}(.) = \psi_x(.)(K_x + \lambda I_N)^{-1} Y_N \tag{25}$$

where $K_x$ is the $N \times N$ block-matrix, with each block of the form $K(x_k, x_l)$, $Y_N$ is the vector of all stacked vectors $\phi(\sigma_1), \ldots, \phi(\sigma_N)$, and $\psi_x$ is the matrix composed of $[K(., x_1), \ldots, K(., x_N)]$. In all our experiments, we used a decomposable gaussian kernel $K(x, y) = \exp(-\gamma \|x - y\|^2) I_m$. The bandwith $\gamma$ and the regularization parameter $\lambda$ were chosen in the set $\{10^{-i}, 5 \cdot 10^{-i}\}$ for $i \in 0, \ldots, 5$ during the gridsearch cross-validation steps. For the k-Nearest Neighbors experiments, we used the euclidean distance and the neighborhood size was chosen in the set $\{1, 2, 3, 4, 5, 8, 10, 15, 20, 30, 50\}$.

**Experimental results.** We report additional results in terms of rescaled Hamming distance $(d_{H_K}(\sigma, \sigma') = \frac{d_H(\sigma, \sigma')}{K^2}$ ) on the datasets presented in the paper and in terms of Kendall's $\tau$ coefficient on other datasets. All the results have been obtained in the same experimental conditions: ten folds cross-validation are repeated five times with the parameters tuned in a five folds inner cross-validation. The results presented in Table 3 correspond to the mean normalized Hamming distance between the prediction and the ground truth (lower is better). Whereas Hamming based embeddings led to very low results on the task measured using the Kendall's $\tau$ coefficient, they outperform other embeddings for the Hamming distance minimization problem as expected.

Table 3: rescaled Hamming distance

|  | authorship | glass | iris | vehicle | vowel | wine |
|---|---|---|---|---|---|---|
| kNN Kemeny | 0.05±0.01 | 0.07±0.02 | 0.04±0.03 | 0.08±0.01 | 0.07±0.01 | 0.04±0.03 |
| kNN Lehmer | 0.05±0.01 | 0.08±0.02 | 0.03±0.03 | 0.10±0.01 | 0.10±0.01 | 0.04±0.03 |
| kNN Hamming | 0.05±0.01 | 0.08±0.02 | 0.03±0.03 | 0.08±0.02 | 0.07±0.01 | 0.04±0.03 |
| ridge Kemeny | 0.06±0.01 | 0.08±0.03 | 0.04±0.03 | 0.08±0.01 | 0.08±0.01 | 0.04±0.03 |
| ridge Lehmer | 0.05±0.01 | 0.09±0.03 | **0.02**±0.02 | 0.10±0.01 | 0.08±0.01 | 0.09±0.04 |
| ridge Hamming | **0.04**±0.01 | **0.06**±0.02 | **0.02**±0.02 | **0.07**±0.01 | **0.05**±0.01 | **0.04**±0.02 |

In Table (4), we show that Lehmer and Hamming based embeddings stay competitive on other standard benchmark datasets. The Ridge results have not been reported due to scalability issues as the number of inputs elements and the output space size grow.

Table 4: Kendall's $\tau$ coefficient on additional datasets

|  | bodyfat | calhousing | cpu-small | pendigits | segment | wisconsin | fried | sushi |
|---|---|---|---|---|---|---|---|---|
| kNN Lehmer | **0.23**±0.01 | 0.22±0.01 | 0.40±0.01 | **0.94**±0.00 | 0.95±0.01 | **0.49**±0.00 | 0.85±0.02 | 0.17±0.01 |
| kNN Kemeny | **0.23**±0.06 | 0.33±0.01 | **0.51**±0.00 | **0.94**±0.00 | 0.95±0.01 | **0.49**±0.04 | 0.89±0.00 | 0.31±0.01 |
| Cheng PL | **0.23** | 0.33 | 0.50 | **0.94** | 0.95 | 0.48 | 0.89 | **0.32** |
| Zhou RF | 0.185 | **0.37** | **0.51** | **0.94** | **0.96** | 0.48 | **0.93** | – |

On the sushi dataset Kamishima et al. (2010), we additionally tested our approach Ridge Kemeny which obtained the same results as Cheng PL (**0.32** Kendall's $\tau$).