[Reviews · NeurIPS 2018]

Reviewer 1



The paper develops methods for the label ranking problem through structured output learning, where the rankings are embedded to Hilbert spaces in such a way that the inner products of two rankings have direct correspondence to particular loss functions for rankings. With these embeddings, one can learn rankings efficiently through, e.g. vector-valued kernel ridge regression. The authors develop algorithms for the preimage problems arising from the use of the different embeddings. In particular, they propose a new Lehmer embedding that has a fast preimage algorithm as well as extendibility to partial rankings. The generalization performance and algorithmic complexity is further analyzed. In numerical experiments, the methods work well but the improvement over previous state-of-the-art is rather small in general. Update: I think the author’s response addresses the criticism posed by the reviewer’s well, in particular I appreciate the progress made in the theoretical side, even if the experimental results are not earth-shattering.

Reviewer 2



This paper presents an interesting approach to the label ranking problem, by first casting it as a Structured Prediction problem that can be optimized using a surrogate least square methodology, and then demonstrating an embedding representation that captures a couple of common ranking loss functions -- most notable being the Kendall-Tau distance. Overall I liked the paper and found a decent mix of method, theory and experiments (though I would have liked to see more convincing experimentation as further detailed below). In particular I liked the demonstration of the Kendall tau distance and Hamming distances to be representable in this embedding formulation/ That said I had a few concerns with this work as well: - Specifically the empirical results were not very convincing. While this may not have been a problem for a theory-first paper, part of the appeal of an approach like this it is supposed to work in practice. Unfortunately with the current (some what limited) set of experiments I am not entirely convinced. For example: This only looked at a couple of very specific (and not particularly common loss functions) with the evals only measuring Kendall Tau. Instead metrics like MAP and NDCG are the more standard metrics typically used these days. Absence of any of those metrics or other standard metrics was quite concerning. Coupled with the result that indicated that optimizing the specific metric does not always lead to the best results on other metrics is why more eval metrics would have helped paint a clearer picture. Furthermore as the authors themselves point out: top-k rankings are a primary focus in most applications and towards this end the paper does not exactly provide a solution for this more practical problems or even discuss variants to tackle it. Furthermore there are no metrics or evals either on this important setting. The datasets used in these evals also are not the most common or standard from the IR perspective and I would have liked to see more standard ranking datasets be used for the evaluation. Lastly I would have liked to see more competitive baselines that capture performance of the different previous works in this field that may not be label-ranking based. This may help contrast this approach versus other methodologies. Would also be great to understand running time for the different methods compared. ---- POST-AUTHOR FEEDBACK EDIT: Thank you to the authors for their response. I appreciate you taking time to clarify some of my concerns. Regarding my main concern about the evaluation: Unfortunately I don't think I felt that was adequately addressed. I shoulder some of the blame for not not being more detailed so let me try to convey my thoughts in more detail below: - The authors mention that their approach is specific to label ranking and not LTR and other related problems. However I think that reduces the appeal of the paper since label ranking is a far less common problem than LTR with a lot more niche applications. If the sole focus was label ranking then I'd like the authors to have explained what problems can only be tackled by label ranking approaches and not by approaches from LTR or multi-class classification approaches. - In particular the latter (multi-class classification) could easily be used as well to produce a "ranking" over the different labeled class. What I don't see is an evaluation justifying the approach for the niche cases multi-class classification cannot handle as well (say where there is a relationship among the labels). - Lastly when I referred to previous approaches, I was thinking about the works by Thorsten Joachims and collaborators for example. I failed to see any mention of how well something like a structual SVM would do on this problem. I feel the paper would be a lot stronger if it discussed (and ideally compared) against these existing approaches.

Reviewer 3



This paper studies label ranking learning. It proposes embeddings for representing rankings and uses the surrogate least square approach for label ranking. It then solves the pre-image problem to recover the predicted ranking in the original permutation space. In general, the pre-image problem is the bottleneck as recovering the ranking in the original space is factorial. The complexity is dramatically reduced, however, if the ranking embedding exhibits certain structure. In this paper, three embeddings are considered: Kemeny (based on the Kendall distance), Hamming, and Lehmer. It provides statistical guarantees for the learning procedures based on all these three embeddings. I think this is a well-written paper and is a good extension of the existing work on label ranking. The surrogate least square approach used in this paper is, as far as I know, significantly different from the existing work, such as Gaertner and Vembu. Section 2.1. The terminology was misused here. It is more common to refer the ranking without ties as strict order rather than total order. I believe the term full ranking used here corresponds to a total order. Minor thing: an Euclidean space. Line 95. Please format your citation right. Line 167. Maybe it is better to move the footnote number somewhere else, as it is mixed with the superscript. Section 4.4. The section title is “extension to partial and incomplete rankings”, so the readers might expect to see discussions on both for all there embeddings. For Lehmer codes, the discussion on incomplete rankings is missing. As a reference, the interpretation of the missing items in rankings is the same as the one used in the decision tree approach (such as Cheng et al). Theorem 1. It is unclear if the results hold for incomplete and partial rankings. Or does it applies with a minor change to the constants? In any case, please indicates the possible limitation of the results. Section 6. The empirical results met the theoretical expectation. Two minor questions: (a) How are concordant and discordant pairs defined? The same as in Eq. (9)? (b) For Kemeny, as Step 2 is NP-hard, how did you solve it? Brute force or with some approximation? -- I have read the author's rebuttal. One comment regarding the feedback to Reviewer 1 & 2: While I think the synthetic data serve the purpose of the paper, I feel I should point out that real-world label ranking data do exist. Sushi data, e.g., is widely used http://www.kamishima.net/sushi/.